# Impact of Sex and Exercise on Femoral Artery Function: More Favorable Adaptation in Male Rats

**DOI:** 10.3390/life13030778

**Published:** 2023-03-13

**Authors:** Márton Vezér, Attila Jósvai, Bálint Bányai, Nándor Ács, Márton Keszthelyi, Eszter Soltész-Katona, Mária Szekeres, Attila Oláh, Tamás Radovits, Béla Merkely, Eszter M. Horváth, György L. Nádasy, Marianna Török, Szabolcs Várbíró

**Affiliations:** 1Department of Obstetrics and Gynecology, Semmelweis University, Üllői Street 78/a, 1082 Budapest, Hungary; 2Department of Physiology, Semmelweis University, Tűzoltó Street 37-47, 1085 Budapest, Hungary; 3Department of Neurosurgery, Military Hospital, Róbert Károly Round 44, 1134 Budapest, Hungary; 4Workgroup for Science Management, Doctoral School, Semmelweis University, Üllői Street 22, 1085 Budapest, Hungary; 5Department of Morphology and Physiology, Faculty of Health Sciences, Semmelweis University, Vas Street 17, 1085 Budapest, Hungary; 6Heart and Vascular Center, Semmelweis University, Városmajor Street 68, 1085 Budapest, Hungary

**Keywords:** exercise training, femoral artery, sex differences, immunohistochemistry, vascular adaptation, favorable males

## Abstract

Blood flow increases in arteries of the skeletal muscles involved in active work. Our aim was to investigate the gender differences as a result of adaptation to sport in the femoral arteries. Vascular reactivity and histology of animals were compared following a 12-week swimming training. Animals were divided into sedentary male (MS), trained male (MTr), sedentary female (FS), and trained female (FTr) groups. Isolated femoral artery rings were examined by wire myography. Contraction induced by phenylephrine (Phe) did not differ between the four groups. The contractile ability in the presence of indomethacin (INDO) was decreased in both sedentary groups. However, we found a specific cyclooxygenase-2 (COX-2) role only in FS rats. After exercise training, we observed increased vasoconstriction in both sexes, when nitro-L-arginine methyl ester (L-NAME) was present. The COX-dependent vasoconstriction effect disappeared in MTr animals, and the COX-2-dependent vasoconstriction effect disappeared in FTr ones. Relaxation was reduced significantly, when L-NAME was present in MTr animals compared to in FTr rats. The training was associated with greater endothelial nitric oxide synthase (eNOS) protein expression in males, but not in females. The present study proves that there are gender differences regarding adaptation mechanisms of musculocutaneous arteries to sports training. In males, relaxation reserve capacity was markedly elevated compared to in females.

## 1. Introduction

The beneficial effects of regular exercise can be seen in several body functions of most mammalian species, including humans. The effects are noticeable in the adaptation of the cardiovascular system [1]. In the lipid metabolism, regular exercise participates in the maintenance of the appropriate body weight and body fat percentage [2]. Regular exercise has been proven to reduce the development of metabolic syndrome, osteoporosis, or even depression [3]. Considering the changes in the cardiovascular system, the most important ones include increases in cardiac output, stroke volume, and heart contractility. As a result of regular exercise, the amount of circulating blood rises and the oxygen consumption increase up to 20–40% (VO2 max) [4]. During exercise, blood flow markedly increases in particular skeletal muscles involved in the given activity [5].

Blood vessels adapt to chronic exercise through their structure and through adaptations of the microcirculation, their number and their diameter increases as well as resistance reduction. Wall thickness decreases, so compliance is increased, optimizing the vessels to meet the demand of elevated blood flow [6]. The vasodilator capacity of blood vessels is also markedly increased. The number of capillaries is elevated in the extremities and in skeletal muscles, increasing the efficiency of oxygen delivery [7].

In addition to the generally known adaptation mechanisms described above, the question arises as to what type of adaptation differences there may be due to the different characteristics of the two sexes. There are significant gender differences between men and women regarding body composition and adaptation to exercise: men have longer limbs, and they have more significant muscle mass, a more robust bone structure, and higher aerobic capacity; women, however, demonstrate better results regarding muscle fatigue and regeneration when performing endurance-type exercise [8]. The maximum performance is not uniform in the sexes, and there is an average difference of 10% between men and women. Such anthropological data for untrained persons have remained unaltered since the 1980s [9]. Gender differences regarding performance of trained persons are lowest for swimmers and highest in track and field sports [8].

Our research team has previously published papers regarding gender differences found in different types of vessel segments as a consequence of adaptation to sports training. After 12 weeks of training, we observed sex differences concerning adaptation mechanisms. Regarding coronaries, we found that the arteriolar segments contractibility was increased in females and relaxation capacity was increased in males [10]. Regarding the gracilis arteriole, tangential wall stress and norepinephrine-induced contraction were elevated in female rats compared to in male animals [11]. During the examination of a visceral large artery (renal artery), we found that phenylephrine-induced vasoconstriction was reduced in males, but not in females [12]. The effect of sex hormones on blood vessels is well-known. Nitric oxide (NO) production is increased in the endothelium as an effect of estrogen [13].

Based on our theorem, the sports adaptation of the femoral artery is more complex than simple sex hormone effects, and it must be influenced by several other factors. Our goal was to understand the sports adaptation of the femoral artery, which is the main artery feeding the lower limb, to reveal the exact mechanisms, while taking into account potential sex differences.

## 2. Materials and Methods

### 2.1. Chemicals

Chemicals for our experiments were sourced from Sigma-Aldrich (St. Louis, MO, USA). Vascular measurements were performed on isolated Wistar rat femoral arteries in a normal Krebs-Ringer solution (nKR; NaCl: 119 mmol/L; NaH_2_PO_4_: 1.2 mmol/L; KCl: 4.7 mmol/L; NaHCO_3_: 24 mmol/L; MgSO_4_: 1.17 mmol/L; glucose: 5.05 mmol/L; ED3TA and CaCl_2_: 2.5 mmol/L). The preparation of the reagents was performed on the day when the experiments were performed, ensuring a fresh solution.

### 2.2. Ethical Approval and Animals

During the experiments, we did everything possible to minimize the suffering or inconvenience of our animals. The committee for Animal Care at Semmelweis University approved our protocols for these experiments (license number: PEI/001/2374-4/2015). This was in accordance with European Union regulations (Directive 2010/63/EU) regarding use and care of animals for research purposes.

All animals were treated according to the Guide for the Care and Use of Laboratory Animals, issued by the National Institute of Health (NIH Publication No. 86-23, Revised in 1996).

The ages of both the male and the female Wistar type rats were 20 weeks old at the time when our experiments were performed on the femoral arteries. The animals were provided at a constant temperature of 22–25 °C and a relative humidity of 40–70% (aka. climate controlled conditions), with a light/dark cycle of 12 h. The animals had free access to standard laboratory food and tap water.

### 2.3. Experimental Groups

Following an acclimatization period of seven days, four experimental groups were established: sedentary male (MS, *n* = 20), trained male, (MTr, *n* = 19), sedentary female (FS, *n* = 18), and trained female (FTr, *n* = 19).

As previously described, a long-term (12 week) swimming training regime was implemented for the trained groups (both males and females) (MTr and FTr) [14]. The rats were placed one by one in tap-water-filled basins providing lanes of 45 × 25 × 20 cm-s. The temperature of the water was kept constant (30–32 °C). The slippery walls and the depth of the basin pools prevented the rats from supporting themselves without making an effort to swim [15]. The training program was built up gradually, starting with 15 min of swimming and increasing by an additional 15 min swim time every other day to a maximum of 200 min daily. Training was performed for the duration of a 12-week period with 5 times each week. To reduce potential differences in swim load, the animals in the sedentary group were placed in the water-filled basin pools for five minutes a day.

### 2.4. Myography

Following the 12-week swimming program, intraperitoneal sodium pentobarbital (45 mg/kg; Euthasol, CEVA Santé Animale, Libourne, France) was administered to anesthetize the animals. In order to prevent intravascular thrombosis, heparinized nKR was perfused for a duration of two minutes into the vasculature. Femoral arteries were cut under a Wild M3Z dissection microscope.

Wire myography (originally described by Halpern and Mulvany) is a fairly standardized, widely applied technique nowadays. We have used this method several times before [16,17]. The experiments were performed on carefully prepared vascular rings using a DMT 610 M Wire Myograph system (multi-chamber isometric myograph system; Danish Myo Technology, Aarhus, Denmark). This system provided 4 separate chambers containing 6 ml-s of nKR each for the organs. Temperature was controlled at a constant of 37 °C, while the pH was maintained at 7.4 via a gas mixture of 95% O_2_ and 5% CO_2_ bubbled through the system. LabChart software was used for data collection (ADInstruments, Oxford, UK-Ballagi LTD, Budapest, Hungary).

After isolation, the femoral arteries were cut into a total of 5 equal pieces with a 2 mm in length each. Four prepared vascular rings were placed on the myograph system. A pretension of 10 mN was applied, which was achieved in 1 h through continuous careful elevation of loadin. Following equilibration, a three-minute course of 124 mmol/L K+ (leading to a contraction of 100%) was administered to test the blood vessel contractility and to set the reference value. Phenylephrine after acetylcholine was administered in a concentration of (10^−6^ mol/L) to determine endothelial viability. Contractility was tested via alpha receptor agonist phenylephrine (Phe), and this was administered in a cumulative manner—the concentration was increased stepwise (10^−8^ to 10^−6^ mol/L). Between the use of different types of vasoactive agents, the organ chambers were thoroughly washed for 3 times on each occasion. The vasorelaxation resulting from acetylcholine (Ach) was measured following preconditioning to contraction with phenylephrine (10^−6^ mol/L) (organ chambers were not rinsed between these steps), and increasing concentrations of Ach (10^−8^ to 10^−6^ mol/L) were administered.

This exact protocol was reperformed following a 30 min pretreatment with nitric oxide synthase blocker nitro-L-arginine methyl ester (L-NAME 10^−5^ mol/L), the cyclooxygenase (COX) inhibitor indomethacin (INDO 10^−4^ mol/L), as well as the cyclooxygenase-2 (COX-2) specific inhibitor NS398 (10^−5^ mol/L).

### 2.5. Histological and Immunochemical Examinations

The vascular segment rings were fixed in formalin and then embedded in paraffin; these blocks were cut into five-micrometer sections. Fixed sections were deparaffinized for immunohistochemical staining. The densities of nitro-tyrosine (NT), endothelial nitric oxide synthase protein (eNOS), and cyclooxygenase 2 enzyme protein (COX-2) were all investigated using immunohistochemistry followed by colorimetry. Tissue handling and staining protocols provided by the manufacturers were applied.

A heated citrate buffer (pH = 6) was used when performing the retrieval of the antigens. H_2_O_2_ (3%) was used to counteract endogenously present peroxidase activity. Non-specific primary antibody binding was prevented by using 2.5% normal horse serum (Vector Biolabs, Burlingame, CA, USA). The following primary antibodies were administered: eNOS mouse monoclonal antibody (1:50; Abcam 76198, Cambridge, UK), COX-2 rabbit polyclonal antibody (1:200; Abcam 15191), and NT rabbit polyclonal antibody (1:500; Merck Millipore AB5411). Secondary labeling was performed by using the following IgG antibodies: polyclonal anti-rabbit (for NT, COX-2) and monoclonal anti-mouse (for eNOS) (BA-2001, Vector Biolabs, Burlingame, CA, USA). Results were visualized using 3,3’-diaminobenzidine (DAB) (Vector Laboratories, Burlingame, CA, USA). Hematoxylin QS (Vector Biolabs, Birmingham, CA, USA) was used for background staining.

A Nikon Eclipse Ni-U microscope equipped with a DS-Ri2 camera (Nikon Minato—Tokyo Japan) was used to obtain images of the histological sections at a 10× magnification for eNOS and a 20× magnification for NT and COX2 stains. ImageJ software (National Institutes of Health (NIH), Bethesda, MA, USA) was used to evaluate the results of the immunochemistry by separating the background staining (DAB + Hematoxylin). Separated images were converted to black and white to quantify the extent of staining using non-calibrated optical density (OD) for the intima layers (in the case of eNOS and COX-2 evaluation) and the media layers (in the case of NT).

### 2.6. Statistical Analysis

Graphical presentation and data analysis were performed using GraphPad Prism software (ver. 8. GraphPad Software, Inc., San Diego, CA, USA). In our series, we expressed data as mean ± SEM. Normal distribution was tested with the Shapiro−Wilks method. If distribution proved to be normal, two-way repeated measures ANOVA was performed for analysis of variance. For post hoc testing, Tukey’s tests were implemented. The results of the immunohistochemical examinations were evaluated with the Kruskal−Wallis test, Dunn’s multiple comparison test, the two-way ANOVA, and the Tukey’s post hoc test. A *p* value of <0.05 was accepted as a level of significance. Data are presented as mean ± SEM.

## 3. Results

### 3.1. Contractility of Femoral Arteries

In femoral arteries, contractility was tested with phenylephrine at increasing concentrations; the four experimental groups did not differ significantly (Figure 1).

The functional vascular effects associated with cyclooxygenases and endothelial oxide synthase (eNOS) were explored as follows: Phe-induced contraction, repeated in the presence of the L-NAME (10^−5^ mol/L), INDO (10^−4^ mol/L), and NS398 (10^−5^ mol/L). In the male and female sedentary groups, the presence of INDO decreased Phe-induced contraction significantly (Figure 2A,C). Following the swimming training, an increased Phe-induced contraction in the presence of L-NAME was observed in both the trained groups such new phenomena (Figure 2B,D). As a gender difference, we found that there was a specific COX-2 vasoconstriction effect only in the FS group (Figure 2C). After exercise training, this specific COX-2 inhibition (NS398) effect disappeared in FTr rats (Figure 2D). Moreover, the INDO effect mentioned in the sedentary groups remained in trained female animals, but not in males (Figure 2B,D). L-NAME effects did not demonstrate significant differences when comparing the trained groups.

### 3.2. Relaxation Ability of Femoral Arteries

Relaxation to Ach induced did not differ in a significant way between the studied groups (Figure 3).

As expected, when L-NAME was administered, Ach-induced relaxation decreased significantly in all animal groups (Figure 4). In addition, when L-NAME was administered in the trained male rats group, Ach-dependent relaxation was markedly decreased compared to in the female rats (Figure 4).

### 3.3. Histological Alterations

The optical density (OD) of eNOS protein expressed was increased in MTr groups after the 12-week training. The OD in the MTr group was higher compared with in the FTr group (Figure 5A,B). Examining COX-2 staining after exercise training, no significant differences were found (26.35 ± 5.299, 77.16 ± 9.001, 38.77 ± 8.580, and 32.93 ± 12.130 arbitrary units for the MS, MTr, FS, and FTr groups, respectively (n.s.)). The OD measured with NT staining also did not show differences (0.08 ± 0.003, 0.07 ± 0.007, 0.08 ± 0.004, and 0.07 ± 0.003 arbitrary units for the MS, MTr, FS, and FTr groups, respectively (n.s.)).

A summary of the results of our present investigation can be found in Figure 6.

## 4. Discussion

In our present series, we demonstrated that, by using the wire myography technique and also performing immunochemistry staining, gender affects sport adaptation of the femoral artery in rodents. Substantial gender differences could be found regarding vascular geometry, contractility, relaxation, and pharmacological properties.

In this study, we found no difference between the groups regarding Phe-induced contraction or Ach-induced relaxation when phenylephrine or acetylcholine was present in the organ chambers by itself. However, if specific inhibitors (INDO, NS398, and L-NAME) were added to the organ chambers in addition to phenylephrine or acetylcholine, we found significant differences both in the effect of training and between the sexes.

In sedentary animals from both sexes, the INDO-COX-dependent vasoconstrictive activity—the prostanoid pathway—was a pivotal factor determining the extent of Phe-induced vasoconstriction. This endothelium-dependent vasoconstriction was abolished by exercise in males. It is known from the literature that as a result of the shear force, the balance between endothelium-derived vasoconstrictor (TXA2, 20-Hydroxyeicosatetraenoic acid) and vasodilator substances (PGI2, 11,12-eicosatrienoic acids) shifts towards vasodilation [18]. Furthermore, treatment of aortas with indomethacin restored the impaired endothelium-dependent relaxation in males, suggesting elevation of cyclooxygenase (COX)-derived vasoconstrictors in aged males [19]. However, the research results are not consistent; we may also read opposite findings: in the thoracic aorta of Sprague Dawley rats, it was observed that vasopressin-induced contraction was attenuated in females, but not in males, by the non-selective COX inhibitor [20]. Gender differences in the level of prostaglandins produced by COX-2 in the kidney were previously described in spontaneously hypertensive rats (SHR) [21]. The urine of female SHR had higher PGE2 metabolite and thromboxane B2 levels compared to that of males. Furthermore, the expression of microsomal PGE2 synthase protein was higher in the renal internal medulla in female SHR, while cyclooxygenase-2 (COX-2) expression was found to be markedly increased in the outer renal medulla of this group. Regarding COX-2, parallel to our research results, it has already been reported that in the case of the thoracic aorta in rats, COX-2 contributes to the extent of contraction in female rat aorta segments to vasopressin via the prostanoid pathway, while COX-1 does not demonstrate this effect [22]. Age-dependent alterations were found regarding cerebrovascular activity to vasopressin following a selective blockade of COX-2 only, and this phenomenon was more pronounced in the female group compared to in the males [23]. However, in this latter case, the difference in contraction agonist, linked to different activation pathways, may be the underlying cause of the difference. In our present experiment, the non-specific COX inhibitory effect was abolished by training in males. This was a contraction-enhancing effect in control animals. Therefore, the balance shifted toward vasodilation as a result of training. Indeed, COX-dependent vasoconstriction remained unaltered in trained females, but COX-2-dependent vasoconstriction disappeared. Thus, the balance shifted towards vasodilation in female rats as a result of swimming.

Investigation of effects of eNOS on vascular function was carried out by repeating Phe-induced contraction following the administration of L-NAME. Phe-induced contraction with L-NAME was enhanced in both trained groups, which suggests that in trained animals the strength of Phe-induced contraction is counterbalanced by NO. No gender differences were found regarding the L-NAME effect between the trained groups, and the degree of Phe-induced contraction is counterbalanced by NO release similarly in both exercised groups. Macedo et al. also found that L-NAME significantly potentiates the vasoconstriction response of Phe in exercised rats [24]. Chronic exercise blunts phenylephrine-induced vasoconstriction in isolated rat aorta, probably by increasing NO release via activation of inducible and endothelial NOS. We may speculate that long-term exercise training increases the gene expression of both inducible and endothelial NOS in isolated rat aortic endothelium and endothelial NOS and neuronal NOS in mesenteric arteries [25].

Following the administration of L-NAME, Ach-induced contraction decreased in all four studied groups in our series. This means that relaxation activity was predominantly mediated by NO in all groups. Relaxation to Ach took place primarily via the NO-dependent pathway. Marchio et al. investigated the effect of long-term training on femoral arteries in male New Zealand white rabbits. In accordance with our results, during the relaxation test with Ach, trained and sedentary groups did not demonstrate differences [26]. Male Sprague Dawley rats also showed no alterations in response to Ach in a previous study [27]. By examining the femoral arteries of mini pigs, these authors also did not find differences between trained versus sedentary groups in terms of Ach-induced relaxation ability [28]. However, the literature is not completely unanimous, as there are known results concerning rat abdominal aorta where increased relaxation occurs to induced vasodilation after exercise training [29]. Based on the data from the literature, we can expect increased NO release and/or activity in different types of blood vessels in association with the presence of estrogen in females [30]. Laughlin et al. examined the femoral and brachial arteries of miniature swine. Based on their results, in the case of the brachial artery, males showed greater acetylcholine- and bradykinin-dependent relaxation compared to females. Regarding femoral arteries, opposite results were observed in control sedentary animals [31]. We did not find differences regarding eNOS protein expression when analyzing data from the control groups; however, there may be a difference in function. This difference in function may mean a gender-related difference in the phosphorylation of eNOS or maybe some other component of the cascade. Further, it is known, for example, that the eNOS dimer-monomer ratio is higher in women. The decrease in the dimer-to-monomer ratio may reflect the uncoupling of eNOS and affect the production of reactive oxygen species (ROS). Cattaneo et al. [32], however, have found no sex differences regarding the amount of released ROS from human endothelial cells.

As a gender difference, the role of NO may decrease after exercise in female rats, compared to in males: the relaxation-reducing effect of L-NAME in Ach-induced relaxation is more pronounced in male swimmers than in female ones. The data in the literature are not consistent regarding training and NO-dependent relaxation. Papers report that NO-dependent relaxation increases as a result of exercise only in women [33]. There are known data showing that NO-dependent relaxation increases in both men and women [34]. At the same time, there is also a result known in the literature, according to which there is no gender difference [35]. Further, others found that brachial artery dilatation increases in response to exercise in men, but not in women [36]. A human study of Dietz et al. compared the forearm vasodilator responses of women and men under the influence of compounds promoting the release of nitric oxide, NO donor compounds, and NO-independent mechanisms. Based on their results, blood flow was decreased in women [37]. In a study performed on humans, Nishiyama et al. implemented a new type of approach to examine vascular reactivity between men and women. The flow-mediated dilatation was normalized to the shear rate, with the aim of excluding mathematical distortions. According to their results, endothelium-dependent vascular reactivity in the lower limb, especially in the case of the popliteal artery, is greater in men [38]. Based on the literature data, we may state that there is a greater increase in muscle mass in males as a result of sports. Therefore, the extent of the increase regarding perfusion was found to be greater in male rates than in female rats [39]. This may be due to the fact that training-induced increased eNOS-dependent relaxation in males [40]. Part of the increased relaxation observed in males may be a greater increase in the amount of eNOS, which we could see in our experiment by the increase of eNOS OD in the trained male group. It is important to note that amount of eNOS activity is not constant throughout the vascular system; in the coronary artery microcirculation, the amount of eNOS increases in a nonuniform manner after training, and there are regional differences due to the effects of shear stress and intraluminal pressure [41]. Finally, we would like to highlight the differences by comparing the current femoral artery with our group’s previous investigation on the renal artery: In the case of the renal artery, Phe-induced vasoconstriction was decreased in the sedentary female group. In males, vasoconstriction decreased as a result of training. Similar to our present results, the non-selective Cox effect contributed to the vasoconstriction of both sedentary groups. However, we found a specific Cox2 effect in males, but not in females. After exercise training, the non-selective COX-dependent vasoconstriction remained in males, but not in females. Regarding vasodilatation, the results were the same between the two vessels [12].

A summary of the result of our present investigation can be found in Figure 6. In conclusion, NO release/bioavailability increased as a result of training counteracts vasoconstriction and improved relaxation in femoral arteries. NO release/bioavailability is likely to be more beneficial in males than in females after training.

### Strengths and Limitations of the Paper

Our present series does not include the analysis of signal and enzyme pathways in a more detailed manner, which could have added further explanation to the observed gender differences. Our observations in highly standardized animal experiments, however, revealed important gender differences regarding the cellular and molecular mechanisms of training adaptation of a large limb artery wall, which is hardly a real possibility in humans due to obvious ethical considerations.

## 5. Conclusions

As a result of swim training, the balance between endothelium-derived vasoconstrictor and vasodilator substances shifted towards vasodilation in both male and female animals. In swimming-trained males, NO-dependent relaxation and relaxation reserve capability increased. We found a greater eNOS expression to be the underlying cause. Sex hormones can have a beneficial effect on eNOS, COX, and COX-2 signaling. In males, adaptation to training appeared to offer more benefits. The gender differences observed in training adaptation can help in the design of optimized exercise training programs in the future.

## Figures and Tables

**Figure 1 life-13-00778-f001:**
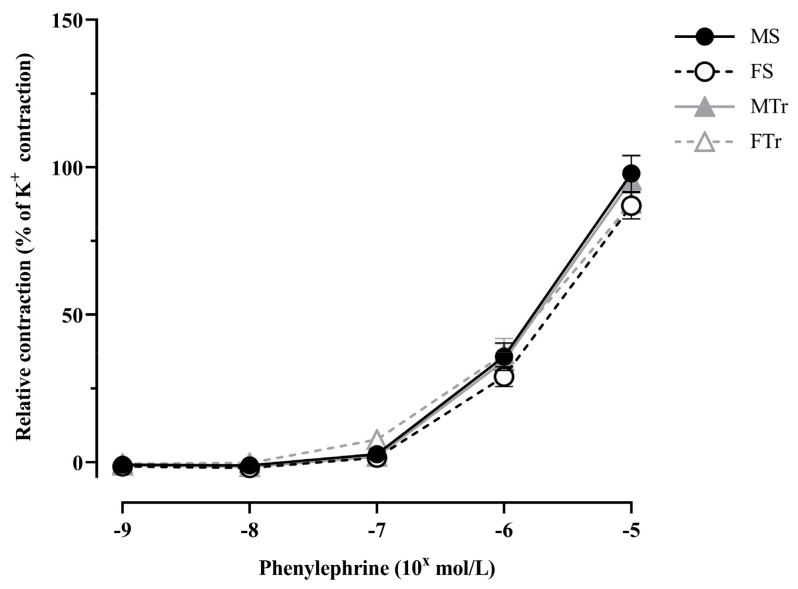
Phenylephrine induced contraction. Data are shown as means ± SEM; *n* = 18–20 in each group; analysis: two-way repeated measures ANOVA; test: the Tukey’s post hoc test. Abbreviations: MS—sedentary male; MTr—trained male; FS—sedentary female; FTr—trained female.

**Figure 2 life-13-00778-f002:**
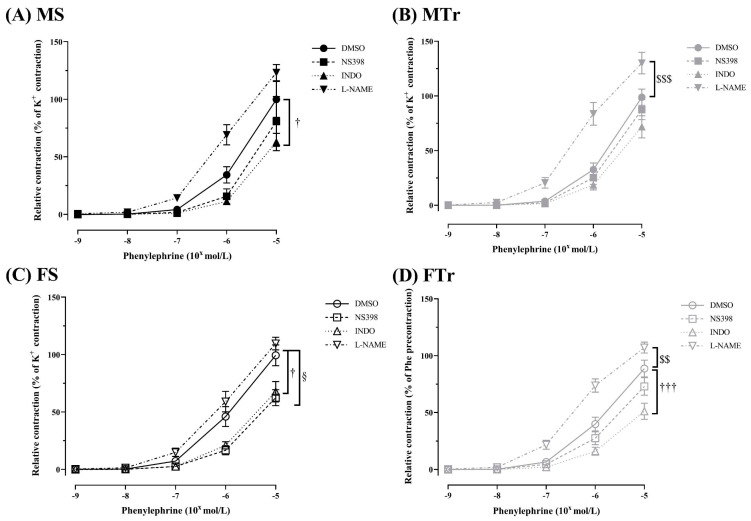
Phenylephrine induced contraction in the presence of NS398, INDO, L-NAME, or DMSO in sedentary male rats (**A**), in trained male rats (**B**), in sedentary female rats (**C**), and in trained female rats (**D**). Data are shown as means ± SEM; *n* = 5–17 in each group; analysis: two-way repeated measures ANOVA; test: the Tukey’s post hoc test. † *p* < 0.05, ††† *p* < 0.001: DMSO vs. INDO; $$ *p* < 0.01, $$$ *p* < 0.001 DMSO vs. L-NAME; § *p* < 0.05 DMSO vs. NS398. Abbreviations: MS—sedentary male; MTr—trained male; FS—sedentary female; FTr—trained female; DMSO—diluted dimethyl-sulfoxide; NS398—the cyclooxygenase-2 specific inhibitor; L-NAME—nitro-L-arginine methyl ester; INDO—indomethacin.

**Figure 3 life-13-00778-f003:**
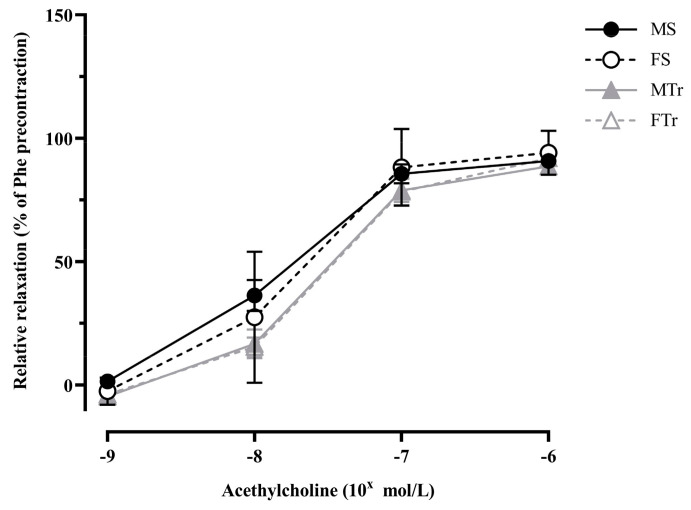
Acetylcholine induced relaxation. Data are shown as means ± SEM; *n* = 15–19 in each group; analysis: two-way repeated measures ANOVA; test: the Tukey’s post hoc test. Abbreviations: MS—sedentary male; MTr—trained male; FS—sedentary female; FTr—trained female.

**Figure 4 life-13-00778-f004:**
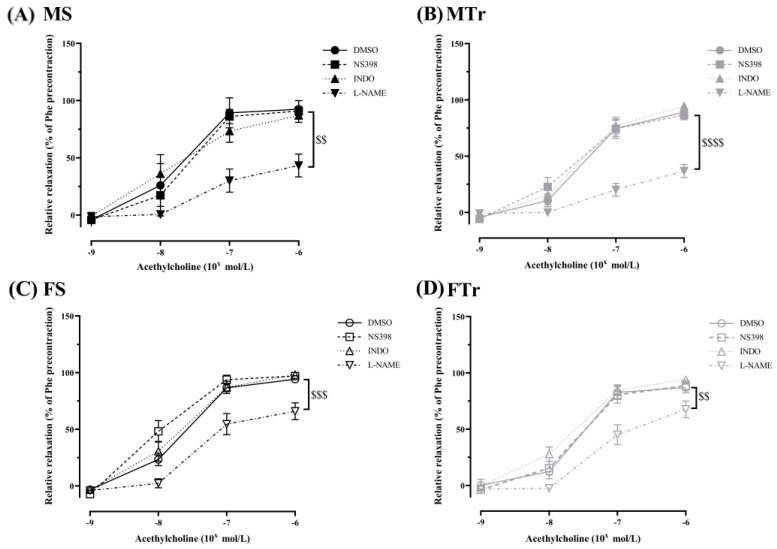
Acetylcholine induced relaxation in the presence of NS398, INDO, L-NAME, or DMSO in sedentary male rats (**A**), in trained male rats (**B**), in sedentary female rats (**C**), and in trained female rats (**D**). Data are shown as means ± SEM; *n* = 5–19 in each group; analysis: two-way repeated measures ANOVA; test: the Tukey’s post hoc test. $$ *p* < 0.01, $$$ *p* < 0.001, $$$$ *p* < 0.0001 DMSO vs. L-NAME. Abbreviations: MS—sedentary male; MTr—trained male; FS—sedentary female; FTr—trained female; DMSO—diluted dimethyl sulfoxide; NS398—the cyclooxygenase-2 specific inhibitor; L-NAME—nitro-L-arginine methyl ester; INDO—indomethacin.

**Figure 5 life-13-00778-f005:**
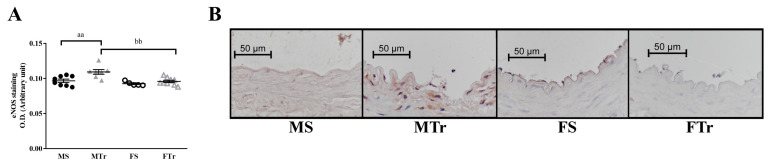
(**A**) Optical density of eNOS labeling in the intimal layer of femoral arteries. Data are presented as individual data points, and lines represent means ± SEM; *n*= 5–10 in each group; analysis: two-way ANOVA; test: the Tukey’s post hoc test. (**B**) Representative images of vessels labeled with an anti-eNOS antibody. Scale bar, 50 µm. aa, *p* < 0.01 MS vs. MTr; bb, *p* < 0.01 MTr vs. FTr. Abbreviations: MS—sedentary male; MTr—trained male; FS—sedentary female; FTr—trained female; OD—optical density.

**Figure 6 life-13-00778-f006:**
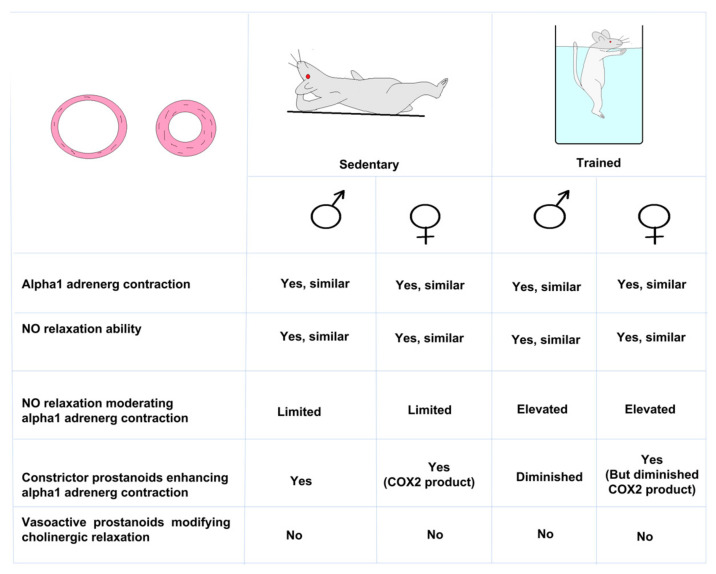
Summary of vascular function changes experienced during a12-week sports adaptation of the femoral artery in a rat model.

## Data Availability

The published article contain all generated and analyzed data from this series.

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
