# Peer review of "Impact of Sex and Exercise on Femoral Artery Function: More Favorable Adaptation in Male Rats"

_life, 2023, doi:10.3390/life13030778_

Round 1

Reviewer 1 Report

In present study, authors reported that femoral artery constriction was COX-dependent in female while it was eNOS-dependent in male rats. Which is very interesting finding. I have some concerns that need to be addressed:

Previously, the same group published similar investigation on renal artery, they found that the eNOS-dependent relaxation became a significant contraction reducing factor in both sexes while In phenylephrine-induced contraction, cyclooxygenase-mediated vasoconstriction mechanisms lost their significance in female rats. Actually either eNOS- or Cox-dependent vascular contraction is upon endothelium. The presented results endothelium acts differentially to adapt exercise-induced stress. If this true, then author should provide evidence at least on differential expression of related genes, such as eNOS and Cox, particularly their active forms, providing kinda  mechanisms.

I am very much surprised that authors did not compare the differential responses of both vessels from two different tissues to exercise. This should add interest or comprehensive understanding on the heterogeneity of vascular contractile at same condition.

How smooth muscle cell functions in response to sport, I mean, SMC-dependent vascular constriction? The contraction with intact or denuded endothelium may provide good model to address such question.

Minors:

The abbreviation should appear when it comes for the first time

The ages of animals were not given.

Author Response

Dear Reviewer1,

Thank you for your thorough revision of our manuscript.

We greatly appreciate both your positive comments and your helpful suggestions.

Our replies are as follows:

English language and style are fine/minor spell check required

A final, thorough linguistic check has been provided by one of the authors with a long stay in the US.  No other alterations have been implemented.

In present study, authors reported that femoral artery constriction was COX-dependent in female while it was eNOS-dependent in male rats. Which is very interesting finding. I have some concerns that need to be addressed:

1.Previously, the same group published similar investigation on renal artery, they found that the eNOS-dependent relaxation became a significant contraction reducing factor in both sexes while In phenylephrine-induced contraction, cyclooxygenase-mediated vasoconstriction mechanisms lost their significance in female rats. Actually either eNOS- or Cox-dependent vascular contraction is upon endothelium. The presented results endothelium acts differentially to adapt exercise-induced stress. If this true, then author should provide evidence at least on differential expression of related genes, such as eNOS and Cox, particularly their active forms, providing kinda mechanisms.

Control of expression and activation of endothelial eNOS and COX are so complex and diverse that there seems to be a possibility for alternative control in chronic physical exercise, under the effect of different agonists, in different vascular territories and in different sexes. We thank the idea to our honored Reviewer, in the prepared review paper we will attempt to draw a signal flow. Much important data, however, is still lacking.

  1. I am very much surprised that authors did not compare the differential responses of both vessels from two different tissues to exercise. This should add interest or comprehensive understanding on the heterogeneity of vascular contractile at same condition.

Thank you very much for the reviewer's suggestion. Previously, we examined the coronary arteries with respect to isolated vessels under similar sports load (PMID: 32051031, 32744049), the gracilis arteries (PMID: 34322036), with pressure myograph, the renalis (PMID: 34995166) and femoral arteries with wire-myograph technique and the coronary artery network with mapping analysis (PMID: 34039432). After the publication of the current and last scientific article, we plan to write a review article in which we summarize the similarities and differences in the case of different types of blood vessels. Our previous investigation, which you also mentioned, was conducted on the renal artery.In accordance with your improvement suggestion, we have indicated the similarities and differences in the manuscript.  

  1. How smooth muscle cell functions in response to sport, I mean, SMC-dependent vascular constriction? The contraction with intact or denuded endothelium may provide good model to address such question.

Thank you very much for the great idea! Unfortunately, we did not have the opportunity to examine the function of SMC cells and the denuded endothelium separately.It is known from the literature that the function of vascular smooth muscle cells changes as a result of regular training. A review written by Koller et al regarding regular aerobic exercise describes in detail the adaptation of the vascular wall and ion channels in the coronary vessels: (PMID: 34358290).  According to the article, increased smooth muscle contraction and altered endothelium-derived contraction factors result in greater vasoconstriction and resting tone.Regular aerobic exercise leads to increased intracellular calcium oscillations, probably through the upregulated calcium-dependent PKC, enhanced voltage-sensitive calcium channels, cGMP-sensitive calcium-dependent chloride channels, and voltage-dependent calcium-dependent potassium channels. The altered modulation of the ion channels - mainly the voltage-dependent Ca2+ channel  - results in an increase in arteriolar tone at rest in trained animals. Regarding studies on Endothel-denuded arterial rings, the study on pigs by Bowles et al should be mentioned (PMID: 9516188). Based on their results, the lack of training in coronary vessels on K+ current characteristics or membrane potential responses in isolated cells suggests that isolated cells lack a factor for enhanced K+ channel activation in arteries from exercised animals, possibly stretch, is absent in isolated cells.  

Minors:

  1. The abbreviation should appear when it comes for the first time.

Thank you very much for your comment, we have corrected our mistake.

  1. The ages of animals were not given.

Thank you very much, you shed light on a really important question. The age of Wistar rats is given in the Materials and Methods chapter. The animals were 8-week-old at the start of the training program.

We would like to thank our Reviewer the careful and detailed overview and useful advices.

We hope that the revised manuscript will be acceptable for publication in Your highly esteemed Journal.

Kind regards, Márton Vezér, Marianna Török and Szabolcs Várbíró

Reviewer 2 Report

1.In page 1, line 30-31

The senrence "Animals were divided into four groups: male-sed-30 entary, male-trained, female-sedentary, and female-trained" is a scientific sentence and its not proper for the part "simple summary"

2. In page 1 line 31-36

Please remark your findings in the part" simple summary" in the form of simple sentence not highly specialized ones.

3. page 1, part" simple summary"

I have a question about this part. Can non-scientific reader, comprehend this part easily and understand the whole purpose of the manuscript?

4. In the sentence "After exercise, the vascular contractility and relaxation differed between the sexes"

Vascular contractility and relaxation need to 

be transformed into more easy-understanding words.

5.In page 1, line 25-36

In the part simple summary, the authors have mentioned some scientific data about their work. Simple summary should be easy to understand for even non-scientist peaple. Please reform it.

6.In page 1 and 2, line 37-50 

Why the authors have mentioned this part in the part " simple summary" it seems to be repetitive (it is similar to line 25-36)

If the line 37-50 belongs to "simple summary" please omit or reconsider it.

7. In page 2, line 55- 58

Please make sure that all sentences in this 

part have relevent scientific references.

 7.Page 2, line 63-69

Why the sentences do not have references?

8. In page 2 

In line 65-70 the authors have written about the effects of regular exercise on body. But in the line 71 they have written about "gender differences between men and women regarding body composition and adaptation to exercise" . This changing of subjects between paragraohes without making a reasonable connection among them disrupts consistency of the text. Please reform this part according to mentioned note.

9. In page 2, line 71-77

Please consider this note that every scientific facts, results and information in the text of the manuscript must have proper related 

reference.

10. In page 2, line 80-90

I am curious that why authors have mentioned this part of the manuscript? 

11. In page 2, line 84 and line 90

Why authors have inserted multiple references in this part?

12. In page 3, line 131, part" 2.4. Myography"

I have some questions about this part. Have the authors performed this part according to a previous scientific guidline? Or they have conducted it based on their prior scientific reseaeches? And why the authors have not mentioned this note in this part?

13. In page 4, line 162, part "2.5. Histological 

and Immunochemical examinations"

Is it not better that the authors mention the scientific source of the protocol that is used in this part? Both for histological examinations and immunochemical examinations

14. In page 8, line 264-276

The authors have mentioned their findings in this part but it is better that they omit this part and turn each one of these findings in to a subheading of the part "Discussion" and after that, discuss about each one of them.

15. In page 10, line 355 and 356

Please reform multiple and middle-sentence references in this part

16. I have a question about the part "Discussion" 

Is it not better to summarize your findings in a one or more simple figure(s)? 

This makes your "Discussion" part more comprehendable.

17. Please check reference list carefully (specially titles and journal names)

18. Please reform sentences without reference, sentences with middle-sentence reference and the ones with multiple reference.

Author Response

Dear Reviewer2,

Thank you for your thorough revision of our manuscript.

We greatly appreciate both your positive comments and your helpful suggestions.

Our replies are as follows:

1.In page 1, line 30-31

The sentence "Animals were divided into four groups: male-sed-entary, male-trained, female-sedentary, and female-trained" is a scientific sentence and its not proper for the part "simple summary"

The sentence is really scientific. We've reworded it.

  1. In page 1 line 31-36

Please remark your findings in the part" simple summary" in the form of simple sentence not highly specialized ones.

Thank you very much for the suggestion. The mentioned sentences have been rewritten for easier understanding.

  1. page 1, part" simple summary"

I have a question about this part. Can non-scientific reader, comprehend this part easily and understand the whole purpose of the manuscript?

We have rewritten it according to your suggestions. We asked non-scientific readers living in our environment for interpretation, so we are confident that the simple summary section is understandable for everyone.

  1. In the sentence "After exercise, the vascular contractility and relaxation differed between the sexes” Vascular contractility and relaxation need to be transformed into more easy-understanding words.

The quoted sentence has been rewritten to make it easier to understand.

5.In page 1, line 25-36

In the part simple summary, the authors have mentioned some scientific data about their work. Simple summary should be easy to understand for even non-scientist peaple. Please reform it. 

We tried to make the sentence easier to understand.

6.In page 1 and 2, line 37-50 

Why the authors have mentioned this part in the part " simple summary" it seems to be repetitive (it is similar to line 25-36). If the line 37-50 belongs to "simple summary" please omit or reconsider it.

You are right. It was really confusing. During the correction, we indicated what a simple summary is and where the abstract chapter of the manuscript begins.

  1. In page 2, line 55- 58 Please make sure that all sentences in this part have relevent scientific references.

Thank you very much for your comment, we have corrected our mistake.

  1. Page 2, line 63-69

Why the sentences do not have references?

References have been inserted.

  1. In page 2 

In line 65-70 the authors have written about the effects of regular exercise on body. But in the line 71 they have written about "gender differences between men and women regarding body composition and adaptation to exercise". This changing of subjects between paragraohes without making a reasonable connection among them disrupts consistency of the text. Please reform this part according to mentioned note.

Thank you for your improvement suggestion, we have corrected it.

  1. In page 2, line 71-77

Please consider this note that every scientific facts, results and information in the text of the manuscript must have proper related reference.

Thank you very much for the improving and important suggestion. We have modified it accordingly.

  1. In page 2, line 80-90

I am curious that why authors have mentioned this part of the manuscript? 

We would like to show that we have previously performed similar sports adaptation experiments on other types of blood vessels. We think it is interesting that the adaptation of different blood vessels is different. If you think this information is not relevant, we will of course remove it from the manuscript.

  1. In page 2, line 84 and line 90

Why authors have inserted multiple references in this part?

We fixed it according to your request.

  1. In page 3, line 131, part" 2.4. Myography"

I have some questions about this part. Have the authors performed this part according to a previous scientific guidline? Or they have conducted it based on their prior scientific reseaeches? And why the authors have not mentioned this note in this part?

We have made the necessary addition.

Wire myography (originally described by Halpern and Mulvany) is a fairly standardized, widely applied technique nowadays. Slight modifications we used can be found in

 Horvath, B., Orsy, P., Benyo, Z., 2005. Endothelial NOS-mediated relaxations of isolated  thoracic aorta of the C57BL/6J mouse: a methodological study. J. Cardiovasc. Pharmacol. 45, 225–231.

Szekeres M, Nádasy GL, Turu G, Soltész-Katona E, Benyó Z, Offermanns S, Ruisanchez É, Szabó E, Takáts Z, Bátkai S, Tóth ZE, Hunyady L. Endocannabinoid-mediated modulation of Gq/11 protein-coupled receptor  signaling-induced vasoconstriction and hypertension. Mol Cell Endocrinol.2015;403:46-56.

  1. In page 4, line 162, part "2.5. Histological and Immunochemical examinations"

Is it not better that the authors mention the scientific source of the protocol that is used in this part? Both for histological examinations and immunochemical examinations

Thank you very much for drawing attention to the deficiency!

  1. In page 8, line 264-276

The authors have mentioned their findings in this part but it is better that they omit this part and turn each one of these findings in to a subheading of the part "Discussion" and after that, discuss about each one of them.

We modified it according to the reviewer's request.

  1. In page 10, line 355 and 356

Please reform multiple and middle-sentence references in this part .

We have reformed the references.

  1. I have a question about the part "Discussion"

Is it not better to summarize your findings in a one or more simple figure(s)?

This makes your "Discussion" part more comprehendable.

Figure 6. Summary of vascular function changes experienced during 12-week sports adaptation of the femoral artery in a rat model.

We hope that our figure presenting the results summarizes the essential changes well.

  1. Please check reference list carefully (specially titles and journal names)

Thank you for the comment. We checked it according to your request.

  1. Please reform sentences without reference, sentences with middle-sentence reference and the ones with multiple reference.

Thank you very much. The error has been corrected. 

We would like to thank our Reviewer the careful and detailed overview and useful advices.

We hope that the revised manuscript will be acceptable for publication in Your highly esteemed Journal.

Kind regards, Márton Vezér, Marianna Török and Szabolcs Várbíró

Reviewer 3 Report

The aim of the work was to determine how prolonged training will affect the contractile properties of the femoral artery in male and female rats. From the data presented, it can be seen that in trained males (MTr) and females (FTr), under conditions of inhibition of NO synthesis under the action of L-NAME, the contraction force significantly increases in response to Phe. Using immunostaining, it was shown that the content of eNOS was increased in the MTr group. The authors showed that the COX-2 inhibitor NS398 reduced femoral artery contraction in untrained female rats. The results obtained are reliable. The conclusions are justified.

Comments

The authors can be recommended to give the average values of the force of contraction of the rings of the femoral artery in response to KCl in groups of animals.

It was shown that in the groups of trained rats the role of eNOS in contraction regulation significantly increased. At the same time, the relaxation caused by Ach does not change. How the authors explain this fact? Ach-induced relaxation is partially preserved in the presence of L-NAME. How can this be explained? What is the mechanism of NO-independent relaxation.

Figure 1 shows that the force of contraction in response to Phe in the FTr and FS groups is exactly the same. However, when comparing the force of contraction in the presence of DMSO in these groups (Fig. 2), it can be seen that the contraction is somewhat weaker in the FTr group. Can this be explained as a DMSO effect?

Author Response

Dear Reviewer3,

Thank you for your thorough revision of our manuscript.

We greatly appreciate both your positive comments and your helpful suggestions.

Our replies are as follows:

The aim of the work was to determine how prolonged training will affect the contractile properties of the femoral artery in male and female rats. From the data presented, it can be seen that in trained males (MTr) and females (FTr), under conditions of inhibition of NO synthesis under the action of L-NAME, the contraction force significantly increases in response to Phe. Using immunostaining, it was shown that the content of eNOS was increased in the MTr group. The authors showed that the COX-2 inhibitor NS398 reduced femoral artery contraction in untrained female rats. The results obtained are reliable. The conclusions are justified.

Comments

The authors can be recommended to give the average values of the force of contraction of the rings of the femoral artery in response to KCl in groups of animals.

Thank you very much for your comment!We provide you with the absolute values ​​of KCl and their average per group. The absolute value of KCl is the difference between the contraction caused by KCl and the preload.

MS: 21,0914606 mN

MTr: 21,1633732 mN

FS: 21,0785676

FTr: 21,5959495

There was no significant difference between the groups in the contraction caused by KCl.

It was shown that in the groups of trained rats the role of eNOS in contraction regulation significantly increased. At the same time, the relaxation caused by Ach does not change. How the authors explain this fact? Ach-induced relaxation is partially preserved in the presence of L-NAME. How can this be explained? What is the mechanism of NO-independent relaxation.

eNOS activity will be dependent both on eNOS expression (number of molecules present) and their controlled activity. There are different control pathways. Ach can effect the production of endogenous vasoactive prostanoids, too.

Figure 1 shows that the force of contraction in response to Phe in the FTr and FS groups is exactly the same. However, when comparing the force of contraction in the presence of DMSO in these groups (Fig. 2), it can be seen that the contraction is somewhat weaker in the FTr group. Can this be explained as a DMSO effect?

Thank you very much for Reviewer’s comment. Statistically, there was no significant difference between the FS and FTr groups as a result of DMSO treatments. DMSO has no effect on blood vessel contractility during in vitro experiments. However, it can inhibit the proliferation of vascular endothelial cells in the G1 phase. PMID: 28396834

We would like to thank our Reviewer the careful and detailed overview and useful advices.

We hope that the revised manuscript will be acceptable for publication in Your highly esteemed Journal.

Kind regards, Márton Vezér, Marianna Török and Szabolcs Várbíró

Round 2

Reviewer 1 Report

I don't have any further concerns

Reviewer 2 Report

I dont have more comment. Thank you for considering my suggestions.